# Electronic transport driven by collective light-matter coupled states in a quantum device

Francesco Pisani [1], Djamal Gacemi[1], Angela Vasanelli[1], Lianhe Li [2], Alexander Giles Davies [2], Edmund Linfield [2], Carlo Sirtori [1] & Yanko Todorov [1]

In the majority of optoelectronic devices, emission and absorption of light are considered as perturbative phenomena. Recently, a regime of highly non-perturbative interaction, ultra-strong light-matter coupling, has attracted considerable attention, as it has led to changes in the fundamental properties of materials such as electrical conductivity, rate of chemical reactions, topological order, and non-linear susceptibility. Here, we explore a quantum infrared detector operating in the ultra-strong light-matter coupling regime driven by collective electronic excitations, where the renormalized polariton states are strongly detuned from the bare electronic transitions. Our experiments are corroborated by microscopic quantum theory that solves the problem of calculating the fermionic transport in the presence of strong collective electronic effects. These findings open a new way of conceiving optoelectronic devices based on the coherent interaction between electrons and photons allowing, for example, the optimization of quantum cascade detectors operating in the regime of strongly non-perturbative coupling with light.

The light-matter interaction process is not an intrinsic property of a quantum system, but is strongly dependent on its electromagnetic environment[1,2]. The emission and absorption processes can be strongly enhanced in a dense ensemble of emitters, which is the essence of Dicke superradiance[3]. These phenomena were shown recently in a solid-state system interacting with infrared photons[4,5]. In this system, comprising a dense two-dimensional electron gas confined in nanometer-sized quantum wells (QWs) in the conduction band of a semiconductor[6], all electrons contribute to a single collective many-body state known as an intersubband plasmon[7,8], where the electronic oscillations are synchronized by the Coulomb interactions[4]. When the two-dimensional plasmon is combined with a resonant microcavity, the cavity photons and the material excitation exchange their energy reversibly at a frequency $\Omega_R$ known as the vacuum Rabi frequency[9–11]. The energy spectrum of the system is then completely changed to yield two light-matter coupled states, intersubband polaritons, separated by an energy $2\hbar\Omega_R$, which have been demonstrated from the THz ($\lambda = 100\,\mu m$)[9] to the MIR ($\lambda = 10\,\mu m$)[10,11] spectral ranges. Collective strong coupling has attracted considerable attention[2–12] as this regime has led to a change in the fundamental properties of materials including the electrical conductivity[13,14], rate of chemical reactions[15,16], topological order[17], and non-linear susceptibility[18], and has enabled new device functionality[19].

The ability to control the epitaxial growth of semiconductor materials allows precise realization of artificial electronic potentials based on tunnel-coupled QWs. This allows, in turn, the single-particle electronic transport to be tailored and thus the extraction of photo-generated electrons to be optimized, a paradigm used for example in quantum cascade detectors (QCDs)[20]. Microcavity-coupled unipolar devices have provided a formidable platform to explore the strong and the ultra-strong coupling regime[21–25]. However, a question that remains to be answered is how the single particle fermionic transport can be

[1]Laboratoire de Physique de l'Ecole Normale Supérieure, ENS, Paris Sciences et Lettres, CNRS, Université de Paris, 24 Rue Lhomond, 75005 Paris, France. [2]School of Electronic and Electrical Engineering, University of Leeds, Leeds LS2 9JT, UK. e-mail: francesco.pisani@phys.ens.fr; yanko.todorov@phys.ens.fr

efficiently coupled with the intersubband polaritons, which are intrinsically bosonic collective states[12]. In the bosonic approach, the matter excitations are modelled as coupled effective harmonic oscillators. While this approach is very efficient to provide the energies of the collective excitations and the light-matter coupled states, it loses track of the dynamic of electronic populations, that are considered as constant parameters[8,12]. Furthermore, in this description the collective states evolve in a sector of the Hilbert space that is orthogonal to the ensemble of single-particle states of the system. Collective effects and strong coupling lead to renormalized light-matter coupled states that are strongly detuned from bare electronic levels responsible for the current flow. This problem is at the heart of the functionalization of semiconductor devices characterized by non-perturbative interaction with light.

Here, we address this problem by experimental and theoretical investigations of semiconductor quantum detectors where a single-particle electronic extractor level is brought into resonance with collective light-matter coupled states. We develop a quantum theory that is free from the bosonisation approach developed previously[2,8,26], and explicitly takes into account the fermionic nature of the carriers, as well as the population dynamics. Our theory quantitatively explains both the magnitude and the spectral features of the photocurrent, and enables understanding of how collective electronic states produce single-particle currents in quantum devices. Our model connects the populations of the single-particle subbands, which drive the electronic transport, to the collective light-induced electronic polarizations through a system of non-linear Bloch-type equations.

## Results

### Microcavity photodetector characterization

The absorbing region of our detector is shown in Fig. 1a and consist of eleven repetitions of four GaAs QWs separated by $Al_{0.35}Ga_{0.65}As$ barriers (the full sequence is provided in the Supporting Information, S.I., section 3). Photons are absorbed in the wide (8.8 nm) QW and promote

electrons from the first to the second level, which have an energy difference of $E_{12} = 108$ meV. Photoexcited electrons cascade through the levels 3, 4 and 5 and are eventually transferred in the next period of the structure. The Fermi level $E_F$ of the structure is 96 meV above level 1, just below the last extraction level. This ensures a high electronic density (~$2.6 \cdot 10^{12}$ cm$^{-2}$) in the main QW, characterized by a plasma energy $\hbar\Omega_P = 70$ meV, which leads to strong collective effects[8]. The absorption and photocurrent of the structure was first characterized in a multipass and mesa configuration (Fig. 1d, e): the experimental results (Fig. 1b) show that the absorption spectrum (blue curve) is significantly shifted from the single particle transition $E_{21} = 108$ meV (dashed line). Indeed, the peak corresponds to the collective electron excitation of an intersubband plasmon[8] at the energy $\widetilde{E}_{12} = \sqrt{E_{12}^2 + (\hbar\omega_p)^2} \simeq 130$ meV. The measured photocurrent is also shifted with respect to the absorption spectrum, strongly pushed towards the extractor level transition 1→3: $E_{31} = 145$ meV. In the Fig. 1c we present the results of our analytical model (see below), which are in excellent agreement with the experimental data. The asymmetry of the photocurrent spectrum is the first experimental signature of the electronic transport being driven by the collective excitation.

In order to achieve the strong coupling regime, the QCD absorbing region is inserted into a resonant double-metal microcavity (Fig. 2a). The microcavity resonance $E_C$ is controlled by the width $w$ of the metallic ridges constituting the top metal layer (see Methods for details). The interaction between the cavity and the intersubband plasmon yields lower (LP) and upper (UP) polariton states, evidenced in reflectivity spectra (Fig. 2b and methods). From the data we measure a Rabi splitting $2\hbar\Omega_R = 22$ meV at 80 K, with is 17% of the electronic excitation transition, confirming the strong coupling regime. The calculated absorption efficiency of the cavity-coupled QWs, $\eta(\hbar\omega)$, is also provided (Fig. 2c). This system allows us to tune the energy of the polariton resonances around the extractor level $E_{13}$, providing a robust test platform for our microscopic model. Normalized photocurrent

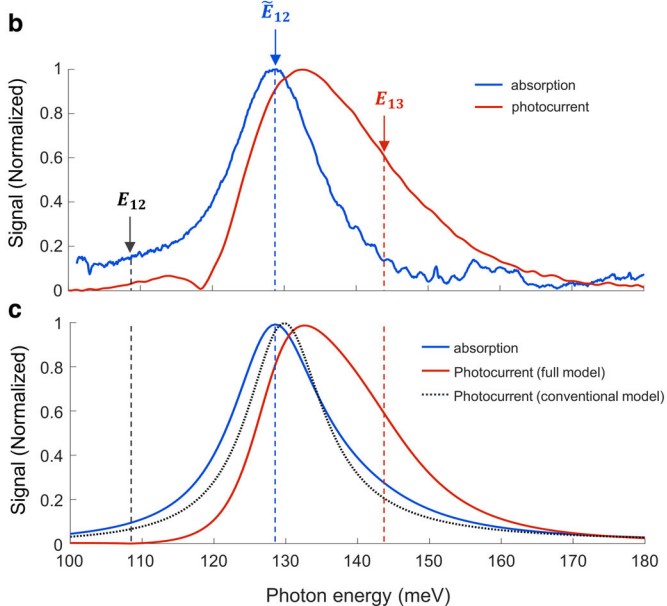

**Fig. 1 | Quantum cascade region design and characterization. a** Schematic diagram of the detector active region. The quantum well on the left is highly-doped, and the optical excitation of the system is a many-body state, an intersubband plasmon, with an energy $\widetilde{E}_{12} = 130$ meV that is higher than the single particle-transition $E_{12} = 108$ meV. The three quantum wells on the right constitute the extractor region for photogenerated electrons and are a single-particle system. **b** Absorption (blue line) and photocurrent (red line) measured respectively in a multipass configuration (**d**) and in a non-resonant cavity (mesa structure, **e**). The energies of the single-particle transition, $E_{21,}$ the collective state $\widetilde{E}_{12}$ and the extractor transition $E_{31}$ are indicated by dashed lines. **c** The measurements in panel **b** are perfectly reproduced by our quantum model. The grey dotted line represents the photocurrent calculated with the conventional **d**etector theory, i.e. $G_H(\hbar\omega) = 1$ (see below). **d** Multipass configuration for the measurement of the absorption in the quantum well. **e** Mesa configuration for the measurement of photocurrent. In both cases the light couples in the sample through a facet polished at a 45° angle.

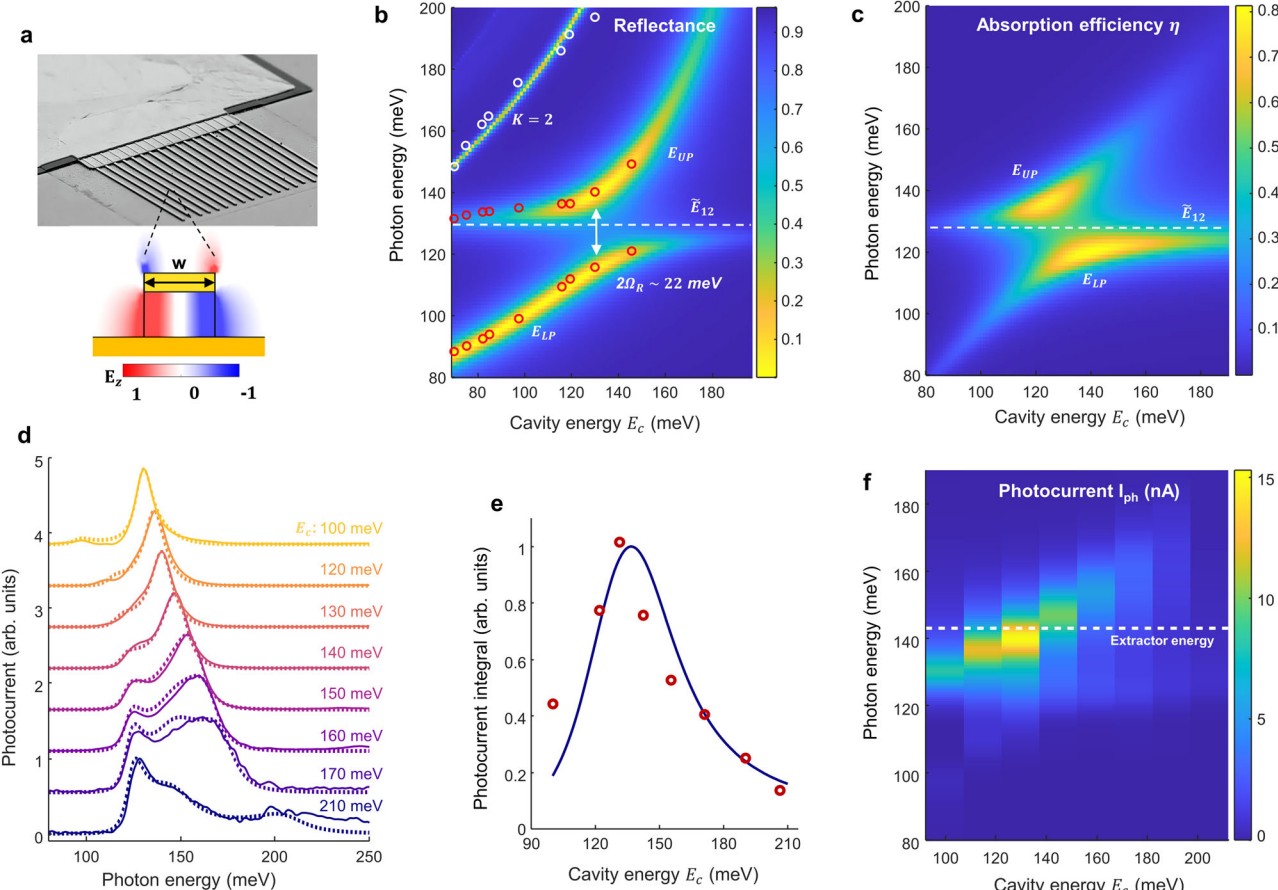

**Fig. 2 | Microcavity photodetector characterization. a** QCD processed into double-metal cavity array. The schematic diagram illustrates the electric field $E_z$ distribution of the fundamental cavity resonance. **b** Simulated reflectivity map from cavity arrays extrapolated from experiments (see Methods and section 2 of S.I.). The microcavity resonance $E_C$ is inversely proportional to the ridge width $E_C \sim 1/w$. Measurements are indicated in circles. Red circles: polariton resonances $E_{UP}$ and $E_{LP}$ arising from the coupling of the fundamental cavity resonance with the intersubband plasmon at an energy $\tilde{E}_{12} = 130$ meV. White circles: second order

cavity resonance (see Methods). **c** Calculated quantum absorption efficiency, i.e. percentage of the incident power absorbed by the active region. **d** Measured (full lines) and computed (dotted lines) photocurrent spectra for structures with various cavity resonance $E_C$ (labels on the spectra). **e** Integrated photocurrent signal from measurements (dots) and model (full line) as a function of $E_C$. **f** Contour plot constructed from the measured photocurrent spectra as a function of the cavity energy and photon energy.

spectra measured for devices with various cavity mode energies $E_c$ are shown in Fig. 2d. Our model (dotted lines in Fig. 2d) reproduces very well the photocurrent spectral features for all cavity energies. It also reproduces the dependence of the photocurrent integral on $E_C$ (Fig. 2e). In Fig. 2f we show a contour plot summarizing all measured photocurrent spectra. While the dispersion of the LP and UP states remain unaltered, their relative intensity is significantly different from the reflectivity and absorption spectra: the photocurrent signal is maximized when the energy of the upper polariton state is aligned with the extractor transition, $E_{UP} = E_{13}$.

## Microscopic model

In order to understand this behavior, we developed a quantum model that employs a microscopic Hamiltonian that takes into account the coupling with microcavity photons, collective electronic effects, and the tunnel-coupling between electronic levels 2 and 3. The Hamiltonian is:

$$\hat{H} = \hat{H}_c + \hat{H}_e + \frac{(\hbar\omega_{P1})^2}{4E_{21}}\hat{P}_{12}^2 + i\hbar\Omega_{R1}(a^\dagger - a)\hat{P}_{12} + \hbar\Omega_t\sum_{\mathbf{k}}\left(c_{2\mathbf{k}}^\dagger c_{3\mathbf{k}} + c_{3\mathbf{k}}^\dagger c_{2\mathbf{k}}\right). \tag{1}$$

The first two terms correspond to the uncoupled systems: $\hat{H}_c = E_c(a^\dagger a + \frac{1}{2})$ is the Hamiltonian of the electromagnetic resonator,

where the operators $a^\dagger/a$ create/destroy a microcavity photon, and the Hamiltonian $\hat{H}_e$ expresses the single-particle electron energies. In the third and fourth terms, $\hat{P}_{12} = \sum_{\mathbf{k}}(c_{1\mathbf{k}}^\dagger c_{2\mathbf{k}} + c_{2\mathbf{k}}^\dagger c_{1\mathbf{k}})$ is an operator that describes the collective electronic polarization between subbands 1 and 2. Here $c_{i\mathbf{k}}^\dagger, c_{i\mathbf{k}}$ create/destroy an electron in subband $i$ with an in-plane momentum $\mathbf{k}$. The third term describes dipole-dipole interactions which lead to plasmonic effects and a blue-shift of the $1 \rightarrow 2$ transition[27]. $\hbar\omega_{P1}$ and $\Omega_{R1} = (\omega_{P1}/2)\sqrt{FE_c/E_{21}}$ are the single electron plasma energy and the light-matter coupling constant, respectively[28]. Here $F = 0.13$ expresses the geometrical overlap between the electron gas and the cavity mode[29]. The fourth term describes the interaction between the $1 \rightarrow 2$ transition and the cavity. The last term of (1), $\hat{H}_T = \hbar\Omega_t\sum_{\mathbf{k}}(c_{2\mathbf{k}}^\dagger c_{3\mathbf{k}} + c_{3\mathbf{k}}^\dagger c_{2\mathbf{k}})$, corresponds to tunneling of electrons between levels 2 and 3. In this picture, the transition $1 \rightarrow 3$ is considered to be a single particle transition with vanishing oscillator strength, while the $1 \rightarrow 2$ transition is strongly renormalized by collective effects and light-matter interaction.

The dynamics of the systems are calculated within a density matrix approach[30], where the Hamiltonian evolution is supplemented with relaxation terms. The system is considered at a temperature $T = 0$ K, with no electrons above the Fermi level in the ground state. An external field, $S_{in}$, drives the cavity, which is coupled with the

electronic populations $N_i = \sum_k \langle c_{ik}^\dagger c_{ik} \rangle$ and coherences $\rho_{ij} = \sum_k \langle c_{ik}^\dagger c_{jk} \rangle$ through the Hamiltonian (1). The case of the mesa structure shown in Fig. 1b is modelled by considering that the driving $S_{in}$ field is applied directly to the electronic system. As explained in the section 1.1 of S.I., this approach generates a set of non-linear equations reminiscent of the semiconductor Bloch equations[31]. For the steady state, the photocurrent can be expressed as $I_{ph} = eN_3/\tau_{34}$, where $N_3$ is the population of the 3rd (extractor) level and $\tau_{34}$ is the 3→4 relaxation time towards the extraction cascade. The average populations, and therefore the steady-state photocurrent, can be expressed from the time-averaged products $\langle E_{cav}\rho_{ij} \rangle$ where the intracavity electric field $E_{cav}$ and electronic coherences $\rho_{ij}$ oscillate at the driving field frequency. In our approach, the problem is solved non-perturbatively accounting for the quadratic and anti-resonant terms of the light-matter coupling Hamiltonian (1). For low incident optical power, the population $N_3$ is proportional to the incident photon flux $|S_{in}|^2$, and we can ultimately express the responsivity of the detector $\mathscr{R}(\hbar\omega) = I_{ph}/\hbar\omega_{12}|S_{in}|^2$ as:

$$\mathscr{R}(\hbar\omega) = \frac{e}{\hbar\omega_{12}} \frac{1}{N_p} G_t G_H(\hbar\omega)\eta(\hbar\omega). \quad (2)$$

Here, the quantity $\eta(\hbar\omega)$ is the total absorption efficiency of the coupled system, that is the fraction of incident photons absorbed by the electronic system (Figs. 1b, 2c). The quantity $G_t = 1/(1 + \tau_t/\tau_{eff})$ is expressed from the characteristic time $\tau_t$ associated with the tunneling process between levels 2 and 3; $\tau_t^{-1} = \frac{2\Omega_t^2 \gamma_{32}}{\omega_{32}^2 + \gamma_{32}^2}$. $\hbar\Omega_t = 4.7$ meV is the tunnel coupling strength (see section 1.2 in the S.I.), $\omega_{32}$ the frequency difference between levels 2 and 3, and $\hbar\gamma_{23} = 2$ meV is the relaxation rate of the coherence $\rho_{23}$. The effective time $\tau_{eff}$ is expressed solely from the population lifetimes (see Eq. (39)in S.I.). Since $0 < G_t < 1$, it can be interpreted as the probability of having an electron transferred to the next period of the quantum cascade once it has been photoexcited. In the conventional QCD theory[32–34], $G_t$ is referred as the extraction probability, $p_e$, and is the only quantity entering the responsivity besides the quantum efficiency and the number of periods $N_p$. $G_t$ is completely independent of the energy $\hbar\omega$ of the absorbed photon but strongly depends on the alignment of the extractor with level 2 (Fig. 3a); it is maximum for a resonant extraction $\hbar\omega_{13} = \hbar\omega_{12}$ and then strongly decays in an almost Lorentzian shape as predicted from the Kazarinov and Suris formula[23–33].

In our model, we have adopted the "Eulerian" picture of the transport in QCD as described in[34], where there is continuity of the current, that is an electron leaving one period is replaced by another electron entering from a contact or the adjacent period. In the expression of the current $I_{ph} = eN_3/\tau_{34}$, $N_3$ is the population of a single period of the QCD. However, the absorption efficiency $\eta(\hbar\omega)$ entering Eq. (2) is the one for the whole absorbing region. Our model described in section 1.1 of the Supplementary material provides the total population of all extractor levels, $N_3 N_p$ which in linear regime is proportional to the total absorption $\eta(\hbar\omega)$. Hence in Eq. (2) we introduce the factor $1/N_p$; in the conventional QCD theory the detector gain, that is the number of electrons circulating in the read-out circuit for each absorbed photon, is thus provided by $G_t/N_p$[32–34].

## Coherent gain $G_H(\hbar\omega)$

A qualitative difference of our microscopic quantum model is the appearance of the additional term $G_H(\hbar\omega)$. This function strongly affects the spectral shape of the photocurrent, and takes into account the effects of electronic transport. We show further that it can describe the "electronic filter" that was phenomenologically introduced in previous works[24–36]. From Eq. (2) the product $G_t G_H(\hbar\omega)$ measures the ability of the detector to convert incident photons into a DC photocurrent; the function $G_H(\hbar\omega)$ (see Eq. (47) in S.I.) ultimately provides a quantitative link between the light-matter-coupled and collective states and the fermionic transport in our system, prescribing the energy alignment of the electronic extractor with the collective states.

In Fig. 3b we plot $G_H(\hbar\omega)$ as a function of photon energy for different values of the extractor transition energy $E_{31} = \hbar\omega_{13}$ from being almost at resonance with the single-particle transition $E_{21} = \hbar\omega_{12}$ (110 meV) to the strongly detuned case ($E_{31} = 160$ meV). In the resonant case, $\hbar\omega_{13} = 110$ meV, which optimizes the fermionic transport, the function $G_H(\hbar\omega)$ is very flat and essentially close to unity. As the extractor transition 1→3 is blue-shifted from the absorbing transition, $G_H(\hbar\omega)$ acquires a spectral shape that peaks slightly above the extractor transition energy $E_{31}$, and quite remarkably, its amplitude increases strongly. Note that the maximum value of $\hbar\omega_{13}$ is naturally limited by the height of the AlGaAs barriers, indeed the level 3 must be sufficiently below the barrier edge to avoid coupling with the continuum. The blueshift of the photocurrent observed in Fig. 1b can thus be explained from the fact that in our design the function $G_H(\hbar\omega)$ has a strong resonance for $\hbar\omega = 145$ meV while the absorption efficiency $\eta(\hbar\omega)$ peaks at $\widetilde{E}_{21} = 130 meV$. We can show (expression (47) from S.I.) that for sufficiently high extraction frequency $\omega_{13} \gg \omega_{12}$, the

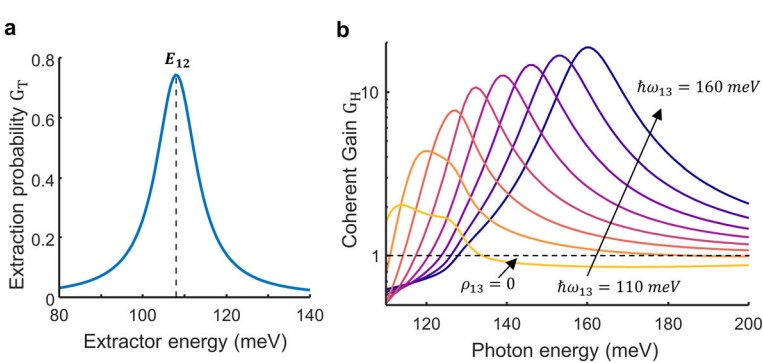

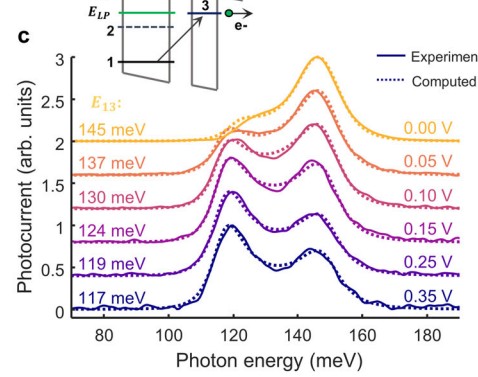

**Fig. 3 | Coherent gain dependence on the extractor energy. a** Extraction probability $G_t$ as a function of the extractor energy. **b** Coherent gain as a function of the photon energy for different energies of the extractor level. The dotted line shows the coherent gain when removing the $\rho_{13}$ coherence **c** Photocurrent spectra measured as a function of an applied bias (labels on the right) for the cavity resonant at $E_C = 140$ meV. The data, continuous lines, are fitted with our model, dotted lines, by changing the extractor energy from 145 meV to 117 meV (labels on the left). The inset shows a simplified scheme of the structure highlighting the 1→3 transition shifted towards the LP energy.

function $G_H(\hbar\omega)$ can be approximated to:

$$G_H(\hbar\omega) \approx 1 + \frac{1}{16\omega_c\tau_{21}} \times \frac{\omega_{21}\omega_{31}\omega_{32}}{\gamma_{21}\gamma_{31}\gamma_{32}} \times \frac{1}{1 + \frac{(\omega-\omega_{31})^2}{\gamma_{31}^2}} \tag{3}$$

Here $\tau_{21}$ is the population relaxation rate from 2 to 1, and $\gamma_{ij}$ are the relaxation rates for the coherences $\rho_{ij}$. Fitting all experimental data (cavities and mesa) provide $\hbar\gamma_{23} = 2$ meV, $\hbar\gamma_{13} = 16$ meV and $\hbar\gamma_{12} = 5$ meV.

The resonant behavior of $G_H(\hbar\omega)$ in Eq. (3) is directly linked to the coherence $\rho_{13}$ between levels 1 and 3. This coherence is induced from the tunneling contribution $\hat{H}_T$ (last term in Eq. (1)) through the equations of motion (see section 1.1 in S.I.). It obeys a forced oscillator equation where the driving terms are proportional to $\Omega_t\rho_{12}$, with $\rho_{12}$ the coherence of the collective states that oscillates at frequency $\tilde{E}_{21}/\hbar$. We can evidence the role of $\rho_{13}$ by artificially removing its effect in the equations by setting a very strong decay rate $\gamma_{13}$, which results in $\rho_{13} \to 0$. For the mesa, the corresponding photocurrent spectra is shown by the dotted line in Fig. 1b, which clearly differs from the experimental results and is almost identical to that of the absorption efficiency $\eta(\hbar\omega)$. The measurements of the polaritonic system in Fig. 2 also clearly show that the resonant behavior of $G_H(\hbar\omega)$ determines the shape and intensity of the photocurrent spectra.

To further illustrate this, in Fig. 3c we change the transition energy $E_{13}$ by applying a bias to the device, as shown in the inset (the measurements were performed on a cavity resonant at 140 meV. See section 4 in S.I. for more voltage-dependent data). The change of relative height between the UP and LP is induced by the bending of the energy bands, lowering the extraction transition energy at positive bias, which becomes resonant with the LP. The continuous lines are obtained by fitting the data with our model from which we established that $E_{13}$ changes from 145 meV at 0 V bias, down to 117 meV at 0.35 V (labels on Fig. 3c).

Note that the absorption efficiency of the collective state is essentially expressed from the coherence $\rho_{12}$: $\eta(\hbar\omega) \propto \langle\rho_{21}E\rangle$, as the transition $1 \to 3$ has vanishing oscillator strength. However, since the electronic transport is relayed through subband 3, the photocurrent is determined by the population $N_3$. The latter is expressed as a linear combination $N_3 \propto G_t(\langle\rho_{21}E\rangle + C\langle\rho_{31}E\rangle)$ ($C$ is a constant, see paragraph 1.3. of S.I.). The increase of $\rho_{13}$ around $\omega \approx \omega_{31}$, $\rho_{31} \propto \rho_{21}/[1 + (\omega - \omega_{31})^2/\gamma_{31}^2]$, yields the resonant contribution in the function $G_H(\hbar\omega)$. Hence, the light-induced population $N_3$ is achieved by two mechanisms: (i) electrons are promoted to the second level by photon absorption and then transferred to level 3 by the tunneling process; and, (ii) electrons are directly transferred from the first to the third level assisted by the collective polarization that appears as a result of the photon absorption.

As seen from Eq. (3), the amplitude of this process is proportional to the quality factors $\omega_{ij}/\gamma_{ij}$ of the three coherences involved, $\rho_{13}$, $\rho_{23}$ and $\rho_{12}$, which explains the strong increase of the maximum of $G_H(\hbar\omega)$ as a function of $E_{31}$. The product of the three frequencies $\omega_{21}\omega_{23}\omega_{31}$ in Eq. (3) bares resemblance with phenomena encountered in non-linear optics[37]; indeed $G_H(\hbar\omega)$ describes the rectification of the fast oscillating optical field and quantum coherences, as the photocurrent generation process is a conversion of an incident AC optical signal into a DC current[38]. This mechanism is due to the intrinsically non-linear character of the Maxwell-Bloch equations in the fermionic system (Eq. (7–24) in the SI), and cannot be recovered from any bosonized models.

### Strong and weak coupling

To examine the results of our model further, in Fig. 4 we show the responsivity simulated for different values of the extractor transition energy $E_{13}$. We compare the case of high (a–d) and low (e–h) doping, which correspond to strong ($\hbar\omega_P = 70$ meV) or weak coupling ($\hbar\omega_P = 7$

meV). Fig. 4d shows the responsivity for $\rho_{13} = 0$ and $G_H(\hbar\omega) = 1$. In this case the responsivity contour plot is identical to the one for the absorption efficiency from Fig. 2c. Indeed, in all the top panels, the QW absorption efficiency $\eta(\hbar\omega)$ (not shown) is identical to the one of Fig. 2c. In contrast, the effect of the dark oscillator associated with $\rho_{13}$ dramatically changes the shape of the photocurrent spectra; we observed efficient photocurrent extraction where the maximum of the function $G_H(\hbar\omega)$ is matched with a maximum of the absorption efficiency $\eta(\hbar\omega)$. The most favorable situation is observed for the extractor matched with the LP, as shown in Fig. 4c, g, thanks to the strong increase of the tunnel contribution $G_t$. On the contrary, in the weak-coupling case, the maximum responsivity is observed for cavities matched with the intersubband transition energy, independently from the position of the extractor, as both the absorption efficiency and the tunnel gain peak at that energy.

In Fig. 4i, j we summarize the maximum responsivity of the QCD, in the weak and strong coupling regimes, as a function of the transition energy $E_{13}$. To construct these curves, we took the absolute maximum of the responsivity for each contour plot shown in the panels above. In the weak coupling regime, the full model (blue line) and the conventional detector theory[32], obtained with $G_H(\hbar\omega) = 1$ (orange line), yield almost identical results: the behavior of the optimal responsivity being governed essentially by the tunnel contribution $G_t$. The results are qualitatively different in the strong coupling regime: the increase of coherent gain $G_H(\hbar\omega)$ with $\hbar\omega_{13}$ and the increase in the absorption efficiency $\eta(\hbar\omega)$ at the UP/LP energy compensates for the decrease of the tunnel gate $G_t$. One can see a strong increase in the responsivity (highlighted by the blue area in Fig. 4j) corresponding to the polariton-induced transport, i.e. when $E_{UP} = \hbar\omega_{13}$ and both $G_H(\hbar\omega)$ and the absorption efficiency $\eta(\hbar\omega)$ are maximized.

## Discussion

Our work thus explains the mechanism for the enhancement of device photoconductivity owing to non-perturbative light-matter coupling and collective effects, and opens further perspectives for optoelectronic devices operating in this regime. So far, we considered a semiclassical limit, where the light field is in a coherent state, $a|\alpha\rangle = \alpha|\alpha\rangle$, and the correlations between the light field and matter polarization are factorized: $\langle(a^\dagger - a)c_{2k}^\dagger c_{1k}\rangle = (\alpha^* - \alpha)\langle c_{2k}^\dagger c_{1k}\rangle$. In the fully quantum version of the model, we can address the problem of vacuum field fluctuations by evolving the system from a vacuum state and considering higher order correlations that appear in the ultra-strong light matter interaction regime. The full quantum version of our microscopic model would also allow thus the study of situations where the intrinsic electronic conductivity is altered by vacuum field fluctuations in a semiconductor quantum detector[12, 39].

An immediate generalization of our approach is to replace the extractor level with an electronic continuum, which will provide a modelling of strongly coupled QWIP structures[40–42]. The description presented here can be generalized to more complex situations where even higher absorption and light-matter coupling strength are achieved, e.g. in the case where the Fermi level lies above several occupied subbands[4]. In that case our approach will provide the most efficient means to energetically align an electronic extractor level with the collective many-body state that is not linked to bare electronic levels. More complex engineering of confined plasmons can be also envisioned[43–45], and our model allows such engineering to be coupled with the single electron transport of the structure. An interesting open question is whether the polariton-induced responsivity observed in Fig. 4j can surpass the maximum for the resonant extraction case. Such detectors would feature very low intrinsic dark currents and higher working temperatures as the extraction channel moves further from the Fermi sea. Other configurations, such as polariton-induced transport in weakly coupled QWs can also be envisioned, a situation reminiscent of the hopping transport in polariton systems based on organic

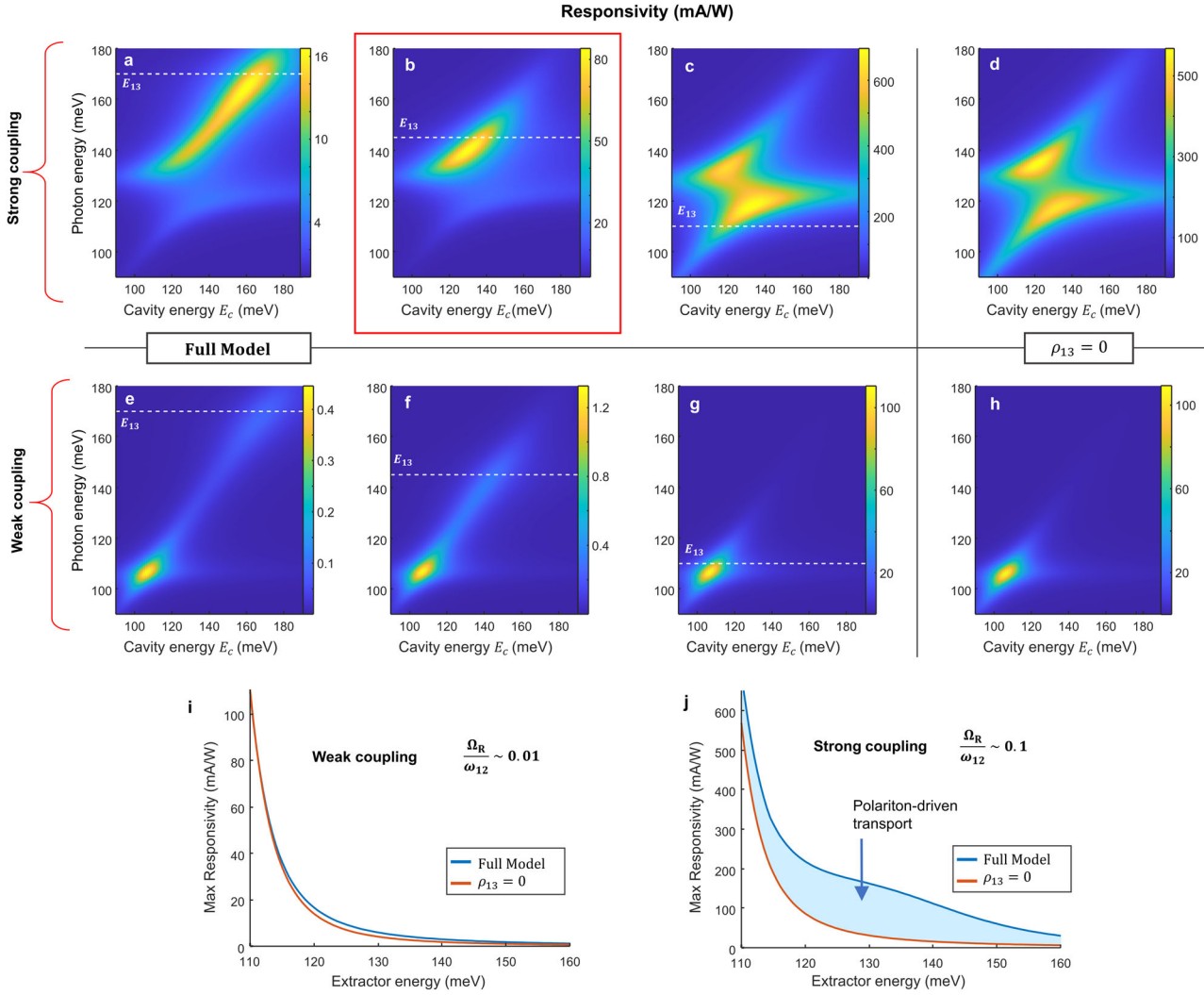

**Fig. 4 | Photodetector responsivity in strong and weak coupling regime.** Calculated responsivity spectra as a function of the cavity and photon energies, for strong (**a**–**d**) $\frac{\Omega_R}{\omega_{12}} = 0.1$ and weak (**e**–**h**) $\frac{\Omega_R}{\omega_{12}} = 0.01$ coupling regimes. The extractor transition $E_{13}$ is set to 160, 145 and 110 meV (dashed line). Fig. 4b, highlighted by the red square, is in excellent agreement with the experimental data from Fig. 2f. In panels (**d**, **h**) we set the coherence $\rho_{13} = 0$ thus removing the effect of the coherent gain (the extractor energy is set at 110 meV). **i**, **j** Maximum responsivity as a function of the extractor energy for weakly (**i**) or strongly (**j**) coupled devices comparing the case in which $\rho_{13}$ is set to zero or not. The enhanced transport driven by the polariton resonance is highlighted by the blue area.

semiconductors[14]. As the microscopic picture developed here allows the interplay between collective electronic effects, electronic transport and electromagnetic properties of metamaterial resonators to be understood, it can successfully be applied for the design of the next generation of quantum detectors operating in the strong and ultra-strong coupling regime.

## Methods

### Multipass absorption

The absorbing region of the QCD detector (see Table S1 in the supporting information for the full description) was epitaxially grown by molecular beam epitaxy onto a 350-µm-thick semi-insulating GaAs substrate. In order to characterize the absorption of the sample in the absence of cavity effects, the substrate was processed in a prism with 45° polished facets. A titanium-gold (Ti/Au) layer was deposited on top of the heterostructure to maximize the field intensity in the active region and ensure internal reflection. The absorption of the structure was measured in a multipass configuration with a FTIR (far-infrared Fourier transform interferometer) and an MCT (mercury cadmium telluride) detector (Fig. 1d). The sample was cooled to 80 K inside a cryostat. To obtain the absorption spectra we measured the ratio of

the transmitted signals for the TM polarization (electric field along the growth direction) and TE polarization (electric field perpendicular to the growth direction); according to the intersubband selection rules, only the TM-polarization is absorbed and the TE one can be used as a normalization spectrum.

### Photocurrent in Mesa structure

For the photocurrent measurements in a non-resonant cavity, the absorbing region was processed into a circular mesa of 200 µm diameter with top palladium germanium (Pd/Ge) and bottom germanium nickel (Ge/Ni) diffused contacts (Fig. 1e). The substrate had a 45° polished facet allowing the TM-polarized light to interact with the active region. The photocurrent generated by the blackbody source of the FTIR was measured with a transimpedance amplifier and a lock-in amplifier, with the sample cooled to 10 K. The measurements are shown in Fig. 1b, red line; no bias was applied to the structure.

### Double-metal microcavity

In order to obtain microcavity-coupled QCDs, as shown in Fig. 2a and Fig. S5a in the S.I., the QCD structure is processed in a double-metal microcavity consisting of a metal ground plane, the semiconductor

containing the quantum heterostructure, and an array of metallic ribbons. In the inset of Fig. 2a we provide the result of a simulation showing the electric field distribution of the resonant electromagnetic mode which is essentially that of a stationary wave vibrating along the ribbon width $w$. The energy of the cavity resonances is provided by $E_{cK} = K\pi\hbar c / n_{eff} w$ where $K > 0$ is an integer, $c$ is the speed of light and $n_{eff}$ is an effective index close to the bulk GaAs index ($n = 3.4$). This configuration allows for very strong electric field confinement in the semiconductor region[46,47] and leads to a strong coupling between the electronic excitation $\tilde{E}_{21}$ and the cavity mode $K = 1$ (in the main text we dropped the index $K$ as we always refer to the first order of resonance). The second order resonance $K = 2$ is also visible in the measurements, but due to the higher frequency its interaction with the intersubband plasmon remains negligible.

### Reflectivity and photocurrent in strong coupling

The strong coupling is evidenced by reflectivity spectra on the sample, that are shown in Fig. 2b and Fig. S5b in the supporting information, where we observe the lower (LP) and upper (UP) polariton resonances. For the reflectivity measurements, we used large area $1 \times 1$ mm$^2$ cavity arrays obtained by a wafer-bonding procedure and electrical lithography to define the ribbons. The light from the FTIR is focused on the sample with a set of mirrors and the reflected signal collected on an MCT detector. As explained previously, the TM polarization was divided by the TE polarization in order to obtain a normalized spectrum. The data (energy of the LP, UP and $K = 2$ mode) superimposed onto the simulated reflectance of Fig. 2b corresponds to the minima in the reflectivity spectra of structures with different ribbon width (see section 4.1 in the supporting information). The simulations were performed with a commercial finite element solver, COMSOL Multiphysics, and the parameters tuned to match the experimental results (section 2 in the S.I.).

For the photodetectors presented in the text, both the bottom and top cavity metal layers are preceded by annealed Pd/Ge ohmic contacts to ensure a good electrical connection with the active region. The latter was also removed everywhere except below the ridges with an inductively coupled plasma (ICP) etching process, a necessary step to avoid spurious mesa-like response generated from the metallic bonding pad. A sketch of the final geometry is reported in Fig. S1 of the S.I.; the area of the photodetectors is ~$100 \times 100$ $\mu$m$^2$ (Fig. S5a in S.I.).

## Data availability

The authors declare that the data supporting the findings of this study are available within the paper and its supplementary information files.

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

## Acknowledgements

We acknowledge funding from the ERC-COG- 863487 "UNIQUE", EPSRC (UK) 'HyperTerahertz' (EP/P021859) and 'TeraCom' (EP/W028921/1) programme grants.

## Author contributions

F.P. and Y.T. wrote the article. C.S. and A.V. provided fruitful discussion to analyze the data and draw conclusions. F.P. fabricated the devices, performed the experiments with the help of D.G. and analyzed the data. He also performed simulations to optimize the detectors and further study the theoretical model developed. L.L., A.G.D and E.L. were responsible for the growth of the heterostructure. Y.T. conceived and developed the theoretical model, supervised and coordinated the project.

## Competing interests

The authors declare no competing interests.
