## [Peer Review File · Nature Communications]

Electronic transport driven by collective light-matter coupled states in a quantum deviceREVIEWER COMMENTS

Reviewer #1 (Remarks to the Author):

In their paper, the authors measure a quantum cascade detector with a large doping in a cavity formed by a metal-metal waveguide shaped into a narrow ridge. In this way, the optical resonance associated with the transverse direction of ridge is brought in resonance with the intersubband system. The authors report measurements of the spectral dependence of the photocurrent and the reflectivity and compare them with a theoretical model.

In general, while the experimental and the theoretical results are clearly presented, the interpretation is confused. Indeed, it is difficult to follow what the central claim is and how it is exactly supported. The introduction of the paper contains many reference to recent work on vacuum field modification of the transport, while neither the theory nor the experiments presented are really relevant to that case. The authors note it themselves: “While we have so far considered a semiclassical limit of the light field, the full quantum version of our microscopic model would also allow the study of situations where the intrinsic electronic conductivity is altered by vacuum field fluctuations in a semiconductor quantum detector”, making it clear that neither the theory nor the experiments are yet relevant for the vacuum fluctuation case.

Measurements of a intersubband detector in the ultrastrong coupling regime, including both reflectivity and photocurrent, were performed first by E. Dupont, H. C. Liu, A. J. SpringThorpe, W. Lai, and M. Extavour “Vacuum-field Rabi splitting in quantum-well infrared photodetectors” Phys. Rev. B 68, 245320 (2003) and which I found surprising not to be cited in this context. The work of Ref 22 also reports photocurrent and reflectivity in detector structures in the strong and ultra-strong light matter coupling regime. The work of Vignerot et al, Appl. Phys. Lett. 114, 131104 (2019), also not cited in the paper, reports the dispersion of the photocurrent and of the reflectivity in such a structure, showing the interplay between the polariton formation and the photocurrent. It is also striking that the polariton splitting, in this last work, is clearly apparent in the photocurrent data while it is not in the work submitted here.

What is very difficult to follow in the submitted work is what the claim is and what is the relation between the claim and the experimental data presented. From the title and the abstract, the key data seems to be in Fig. 4 by the shaded blue area, where the polariton transport is exhibited as the difference between two computed curves, one of which represents a physical situation which is unclear. In the paper, the author say “we can artificially remove the effect of ρ_{13} by setting γ_{13} to infinity”, but doing so does not correspond to a clear experimental situation, as it would represent coupling the upper state to a continuum in which case the meaning of the detuning is lost.

What is also unclear in Fig. 4 is the meaning of the weak or strong coupling, as the light-matter coupling cannot be changed without modifying the doping of the wells. If the authors would want to make a claim, they should compare the situation of the cavity with the one of the mesa and show that, beyond the cavity enhancement that is expected, an additional “polariton transport” is arising and that it corresponds to the experimental measurement.

Reviewer #2 (Remarks to the Author):

In this paper, the authors investigate how the light-matter strongly coupled states are converted to a photocurrent by using a quantum cascade detector (QCD) structure. The quantum well in the QCD structure was very heavily doped. As a result, the intersubband plasmon mode is strongly coupled with the carefully designed cavity mode and forms a strongly coupled state. The electron extractor states of the QCD can be energetically tuned by applying a bias voltage and the photocurrent can be obtained in a spectroscopic manner. The authors observed an enhancement in the responsivity spectrum of the QCD structure when the light-matter coupling is strong. They developed a theory that takes into account the coupling with microcavity photons, collective electronic effects, and tunnel coupling with the extractor states, and clearly showed the coherence between the ground state (1) and the extractor state (3) plays a very important role in the photocurrent generation process, although the oscillator strength between 1 and 3 is negligibly small. This is the process in which electrons are directly transferred from state 1 to state 3 assisted by the collective polarization due to photon absorption. In other words, the enhancement in the responsivity results from the rectification of the fast oscillating optical field and quantum coherences, i.e., the conversion from ac optical fields to a dc current. The authors performed their experiments carefully and, furthermore, developed a quantum theory that nicely explains the observed photocurrent spectra. This paper describes how the strongly coupled light-matter states couple with the electron transport and would become a seminal paper in this field. Therefore, I strongly recommend the publication of this paper in Nature Communications.

Before publication, I have a few very minor questions/comments;

1. The QCD sample used in this study has 11 repetitions of the QCD unit and the quantum wells are heavily doped. In the analysis, the authors assume that, when a voltage is applied to the sample, the electric field is uniform over an entire structure. However, it is known that, when multiple quantum wells are doped, as in the cases of quantum well infrared photodetector structures, so-called high-field domains are formed. This effect strongly modifies the internal field distribution. I wonder how the authors avoid or take into account the effect of high-field domains in their experiments.

2. (very minor comment)

1) line 215: the authors say “at negative bias”. However, it was confusing to me, because I thought the authors control the position of level 3 with respect to the intersubband transition. In this case, I think “at positive bias” seems to be more appropriate.

2) line 96 (typo?): “with”  “which”

Reviewer #3 (Remarks to the Author):

This paper presents an experimental and theoretical study of electronic transport in an infrared detector structure involving ultrastrong light-matter coupling between an intersubband plasmon and an optical resonator mode. Conceiving and fabricating such a device being a remarkable achievement by itself, this also provides an interesting opportunity to investigate mixed light-matter excitation and the impact of resonator-induced detuning of electronic states on the electronic transport.

The theoretical description of the results is based on a microscopic Hamiltonian comprising the electronic states, the resonator mode, the collective electronic polarization, and the interaction between electronic transitions and the cavity. Assuming weak optical excitation, a density matrix approach allows for calculating the stationary photocurrent including analytical expressions.

An interesting feature of this approach consists in the explicit treatment of the resonator-induced coherence between the extractor state and the intersubband transition, which yields extra gain and provides deeper insight in the impact of this non-perturbative light-matter coupling on the photoconduction mechanism.

The paper is well written and describes significant progress in the field of infrared photodetection under ultrastrong, non-perturbative coupling with the resonator mode. The results are convincing and the methodology sound. I recommend publication in Nature Communications with some minor revisions as indicated below.

a) Some words regarding the influence of the number N_p of QCD periods would be important. In particular, the sentence in lines 162-166 regarding the interpretation of G_t is contradictory: If one electron is transferred from one period to the next one, $1/N_p$ electrons will circulate in the external circuit. Therefore, "the number of electrons circulating in the read-out circuit for each absorbed photon" is not the same as "the probability of having an electron transferred to the next period of the quantum cascade once it has been photoexcited". The value of these two terms differs by the factor N_p . Please check.

The small peak value of ca. 4% in Fig. 3a may suggest that the factor $1/N_p$ is already contained implicitly in G_t (in this case, the factor $1/N_p$ should be added to the definition of G_t rather than to eq. (2)).

b) In Fig. 3b, the peak value of G_H is about the same as N_p (presumably $N_p=11$ was chosen because of previous results). Does theory predict such a limit (i.e., that coherence is limited by the size of the device)?

c) Fig. 1b, upper panel: Most likely, the zero of the photocurrent at about 118meV is caused by a sign reversal. This might be attributed to the 1-5 transition leading to transport in the opposite direction. Perhaps the authors could briefly comment on this.

Dear editor and referees,

Here is our answer to the answer to the comments and questions of the referees.

We reported in italic the comments from the referees. The answers follow in normal font. We replied point to point starting from the first referee (R1) and numbered the answers accordingly (A1.1 ...). We added figures and a reference list after the answer of each referee where needed. Please notice that all the modification to the text are also highlighted in green in the revised file.

Sincerely,

Francesco Pisani, Yanko Todorov and the authors.

Referee 1

R1.1. *In general, while the experimental and the theoretical results are clearly presented, the interpretation is confused. Indeed, it is difficult to follow what the central claim is and how it is exactly supported. The introduction of the paper contains many references to recent work on vacuum field modification of the transport, while neither the theory nor the experiments presented are really relevant to that case. The authors note it themselves: “While we have so far considered a semiclassical limit of the light field, the full quantum version of our microscopic model would also allow the study of situations where the intrinsic electronic conductivity is altered by vacuum field fluctuations in a semiconductor quantum detector”, making it clear that neither the theory nor the experiments are yet relevant for the vacuum fluctuation case.*

A1.1. We thank the referee about his remark. Concerning the introduction, we have recalled the broad context of this work, which is the exploration of the ultra-strong light-matter coupling in devices and the associated fascinating phenomena. Vacuum field induced change in the electronic transport are only one of the aspects for ultra-strongly coupled devices among others that we mention. This is indicated in the abstract, as well introductory paragraph: “Collective strong coupling has attracted considerable attention [12-2] as this regime has led to a change in the fundamental properties of materials including the electrical conductivity [13,14], rate of chemical reactions [15,16], topological order [17], and non-linear susceptibility [18], and has enabled new device functionality [19].”

However, we do believe that the approach developed here will permit to tackle the complex problem of evaluating the influence of the vacuum field on the electrical transport. Indeed, as we indicate in page 3 of the Supplementary material:

Page 3: “Equations (9)-(21) derive from Eq. (5), assuming the absence of correlations between the photon field and the quantum well coherences: i.e. $\langle (a^\dagger - a)c_{2k}^\dagger c_{1k} \rangle = (\alpha^ - \alpha)\langle c_{2k}^\dagger c_{1k} \rangle$. We call this condition “semiclassical condition”; it should be revised in the deep ultra-strong coupling where one seeks to establish the link between vacuum field fluctuations and dark current of the detector [4].”*

Therefore, the model described here allows further extension where the above hypothesis of the correlator $\langle (a^\dagger - a)c_{2k}^\dagger c_{1k} \rangle$ is relaxed. The problem of vacuum field fluctuations can be resolved assuming the system is initially in the vacuum state and, for instance, solving the resulting equations of motion, involving the higher order correlators, $\langle (a^\dagger - a)c_{2k}^\dagger c_{1k} \rangle$ as a quantum BBGKY hierarchy problem [<https://doi.org/10.1017/CBO9781139016926>]. To underline this aspect of our work, we have expanded the concluding paragraph of page 8 with the following sentence:

“So far, we considered a semiclassical limit, where the light field is in a coherent state, $a|\alpha\rangle = \alpha|\alpha\rangle$, and the correlations between the light field and matter polarization are factorized: $\langle (a^\dagger - a)c_{2k}^\dagger c_{1k} \rangle = (\alpha^* - \alpha)\langle c_{2k}^\dagger c_{1k} \rangle$. In the fully quantum version of the model, we can address the problem of vacuum field fluctuations by evolving the system from a vacuum state and considering higher order correlations that appear in the ultra-strong light matter interaction regime. The full quantum version of our

microscopic model would also allow thus the study of situations where the intrinsic electronic conductivity is altered by vacuum field fluctuations in a semiconductor quantum detector [**Error! Reference source not found.,Error! Reference source not found.**].”

The objective of the current work. We must note that both in the abstract and the introduction we have stated the objective of our work: Abstract: “Our experiments allowed us to develop a microscopic quantum theory that solves the problem of calculating the fermionic transport in systems where strong collective electronic effects are present.” At the end of the 2nd paragraph we state: “However, a question that remains to be answered is how the single particle fermionic transport can be efficiently coupled with the intersubband polaritons, which are intrinsically bosonic collective states [12]. This problem is at the heart of the functionalization of semiconductor devices characterized by non-perturbative interaction with light. Indeed, collective effects and strong coupling lead to renormalized light-matter coupled states that are strongly detuned from bare electronic levels, which are responsible for the current flow.” In the beginning of the 3rd paragraph we state: “Our experiments allowed us to develop a microscopic quantum theory that solves the problem of calculating the fermionic transport in systems where strong collective electronic effects are present.”

In order to comply with the referee’s remark, and display better the objective of our work with previous approaches, we have commented the particularity of the bosonisation approach:

Page 2, end of 1st paragraph: we added the following explanation:

“In the bosonic approach, the matter excitations are modelled as coupled effective harmonic oscillators. While this approach is very efficient to provide the energies of the collective excitations and the light-matter coupled states, it loses track of the dynamic of electronic populations, that are considered as constant parameters [8, 12]. Furthermore, in this description the collective states evolve in a sector of the Hilbert space that is orthogonal to the ensemble of single-particle states of the system. Collective effects and strong coupling lead to renormalized light-matter coupled states that are strongly detuned from bare electronic levels responsible for the current flow. This problem is at the heart of the functionalization of semiconductor devices characterized by non-perturbative interaction with light.”

Page 2, 2nd paragraph, we added an explanation (added text underlined): “We develop a quantum theory that is free from the bosonisation approach developed previously [2,8,26], and explicitly takes into account the fermionic nature of the carriers, as well as the population dynamics.”

R1.2. *Measurements of a intersubband detector in the ultrastrong coupling regime, including both reflectivity and photocurrent, were performed first by E. Dupont, H. C. Liu, A. J. Spring Thorpe, W. Lai, and M. Extavour “Vacuum-field Rabi splitting in quantum-well infrared photodetectors” Phys. Rev. B 68, 245320 (2003) and which I found surprising not to be cited in this context. The work of Ref 22 also reports photocurrent and reflectivity in detector structures in the strong and ultra-strong light matter coupling regime. The work of Vigneron et al, Appl. Phys. Lett. 114, 131104 (2019), also not cited in the paper, reports the dispersion of the photocurrent and of the reflectivity in such a structure, showing the interplay between the polariton formation and the photocurrent. It is also striking that the polariton splitting, in this last work, is clearly apparent in the photocurrent data while it is not in the work submitted here.*

A1.2. Both works cited by the referee concern quantum well infrared photodetectors, QWIPs, where electrons are scattered directly in the continuum of excited states. Here, we were interested in resonant extraction from the polariton and collective states, which can be engineered in QCD detectors, rather than QWIPs. The work of Dupont et al. is an early work where QWIP absorbing medium was coupled with a dispersive waveguide mode. Polariton splitting in cavity-coupled QWIPs were reported prior to the work of Vigneron et al by our group, see <https://aip.scitation.org/doi/full/10.1063/1.4862750>, a reference that was not included for the reasons state above. However, we do believe that our model can

be generalized to that of a QWIP by replacing the resonant level 3 by a continuum of levels. We therefore include the aforementioned references and the following phrase in the main text:

Page 9, concluding paragraph: “An immediate generalization of our approach is to replace the extractor level with an electronic continuum, which will provide a modelling of strongly coupled QWIP structures [40,41,42]”

Note that the work of Vigneron et al. employs a phenomenological filtering function, that was already introduced by Sapienza et al. (Ref [24] in the main text); but no microscopic model was provided there.

Concerning the visibility of the Rabi splitting in our data: the dispersion of the two polariton branches is clearly visible both in the photocurrent and reflectivity spectra in Figure 2b and 2f; furthermore, the two polariton branches acquire similar weight in the photocurrent spectra in the case where the level 3 is brought in resonance with level 2 (see 3c and compare with 4c).

References ... were added to the text.

R1.3. *What is very difficult to follow in the submitted work is what the claim is and what is the relation between the claim and the experimental data presented. From the title and the abstract, the key data seems to be in Fig. 4 by the shaded blue area, where the polariton transport is exhibited as the difference between two computed curves, one of which represents a physical situation which is unclear. In the paper, the author say “we can artificially remove the effect of ρ_{13} by setting γ_{13} to infinity”, but doing so does not correspond to a clear experimental situation, as it would represent coupling the upper state to a continuum in which case the meaning of the detuning is lost.*

A1.3. We disagree with the referee with on that point, as setting γ_{13} to infinity (thus setting ρ_{13} to zero) does not mean replacing level 3 with a continuum. Indeed, in a general 3 subband system with a Hamiltonian $\hat{H} = \sum_{\mathbf{k}} \hbar\omega_{1\mathbf{k}} c_{1\mathbf{k}}^\dagger c_{1\mathbf{k}} + \sum_{\mathbf{k}} \hbar\omega_{2\mathbf{k}} c_{2\mathbf{k}}^\dagger c_{2\mathbf{k}} + \sum_{\mathbf{k}} \hbar\omega_{3\mathbf{k}} c_{3\mathbf{k}}^\dagger c_{3\mathbf{k}}$ the density matrix that we consider, $\rho_{ij} = \sum_{\mathbf{k}} \langle c_{i\mathbf{k}}^\dagger c_{j\mathbf{k}} \rangle$ is written as (we have $N_i = \rho_{ii}$)

$$\begin{bmatrix} N_1 & \rho_{12} & \rho_{13} \\ \rho_{12}^* & N_2 & \rho_{23} \\ \rho_{13}^* & \rho_{23}^* & N_3 \end{bmatrix}$$

We have compared that general case with the case where the off-diagonal element ρ_{13} is set to zero, $\rho_{13} = 0$:

$$\begin{bmatrix} N_1 & \rho_{12} & 0 \\ \rho_{12}^* & N_2 & \rho_{23} \\ 0 & \rho_{23}^* & N_3 \end{bmatrix}$$

which means that we have removed the quantum coherence between the level 1 and level 3. Nevertheless, the level 3 remains perfectly well-defined in its energy position with respect to the other energy levels in the system. In particular we keep a finite quantum coherence ρ_{23} between level 2 and level 3, which yields a peaked behavior of the tunnel contribution G_i (Fig. 3a). In our equations, the rate of excitation of the coherence ρ_{13} appears to be proportional to $1/\gamma_{13}$, therefore it can be removed by setting $\gamma_{13} \rightarrow \infty$, which means that the coherence ρ_{13} never reaches significant values as it is overdamped.

However, we do agree with the referee that reasoning on the damping rate γ_{13} alone induces a confusion. Therefore, we have replaced “ $\gamma_{13} \rightarrow \infty$ ” with $\rho_{13} = 0$ in our Figures and text.

Let us now comment the significance of the shaded area in Fig. 4j. As we explain in the main text setting the coherence $\rho_{13} = 0$, leaves us with a coherent gain $G_H(\omega) = 1$, which lead to the expression for the responsivity

$$\mathcal{R}(\hbar\omega) = \frac{e}{\hbar\omega} \frac{1}{N_p} G_t \eta(\hbar\omega).$$

We thus recovered the responsivity predicted by the standard QCD theory [1] (see also our reply **A3.1**). In that case the tunnel contribution G_t can be identified with the extraction probability, that is the probability to transfer the photoexcited electron from level 2 to level 3 and then to the read-out circuit. In that framework the maximum responsivity is always achieved when the extractor is resonant with the intersubband transition (see Fig. 3a of the main text). Since, in the strong coupling regime the absorption efficiency $\eta(\hbar\omega)$ is maximum at the energies of the polariton states (Figure 2c of the main text), one should expect lower responsivity from the polaritonic branches than what is actually measured (Fig. 4j, orange curve). In that conventional picture the dependence of the responsivity on the position of the extractor level is identical to the case of weak coupling (Fig. 4i), as it is dominated by the behavior of G_t . Instead, we show that the presence of the coherence ρ_{13} in our model (Fig.4j, blue curve) allows for an additional transport channel which appears due to the resonant shape of $G_H(\hbar\omega)$. As a result, it becomes possible to efficiently extract photocurrent from the upper polariton branch. Our work thus explains the mechanism for the enhancement of device photoconductivity owing to non-perturbative light-matter coupling and collective effects, and allows further optimizations through the quantum design of such detectors.

- [1] Delga, Alexandre. "Quantum cascade detectors: A review." Mid-infrared Optoelectronics (2020): 337-377.

R1.4. *What is also unclear in Fig. 4 is the meaning of the weak or strong coupling, as the light-matter coupling cannot be changed without modifying the doping of the wells. If the authors would want to make a claim, they should compare the situation of the cavity with the one of the mesa and show that, beyond the cavity enhancement that is expected, an additional "polariton transport" is arising and that it corresponds to the experimental measurement.*

A1.4. In our manuscript, we compare the single-particle picture employed to describes detectors with a more complete model which includes collective effects and light-matter coupling. In our Hamiltonian, Eq. (1), these effects are provided respectively by the third term, $\frac{(\hbar\omega_{P1})^2}{4E_{21}} \hat{P}_{12}^2$ and the fourth term, $i\hbar\Omega_{R1}(a^\dagger - a)\hat{P}_{12}$. Both these terms increase with the plasma frequency, ω_{P1} , since the Rabi splitting in these systems is expressed as $\Omega_{R1} = (\omega_{P1}/2)\sqrt{FE_c/E_{21}}$. By setting $\omega_{P1} = 0$, we can remove both these effects and we recover the usual theory of detectors. In Figure 4, we can thus compare the spectral features expected in the weak and strong coupling regime. In that Figure, we also observe that the absolute responsivity for highly doped structures is larger, as the number of absorbers is higher, as expected. However, in the strong coupling regime the spectral behavior of the photocurrent and its magnitude now depend on both the design of the quantum structure and the polariton dispersion.

The comparison that is required by the referee (mesa versus cavity) is actually already present in the manuscript: this is the comparison between Figure 1b,c and Figure 2d,e. It can be seen that, in particular, the cavity effect allows achieving photocurrent signal at higher energies (maximum around 145meV, fig. 2e) than the case of the mesa (130meV, Fig.1d). It should be noted that, as we comment in the main text, the case of the mesa is interesting, as already in that case the dipole-dipole interactions create a collective state with an energy $\tilde{E}_{21}=128$ meV that is completely different from the single particle transition energy, $E_{21}=109$ meV. Indeed, in that case, we have $\Omega_{R1} = 0$ ($F = 0$) but the plasma frequency has the same value as in the cavity-coupled case. The corresponding Hamiltonian is still not

trivial as the many-body term $\frac{(\hbar\omega_{P1})^2}{4E_{21}}\hat{P}_{12}^2$ is significant. As shown in Fig. 1 our theory already reproduces the data in that case.

Referee 2

R2.0 "...The authors performed their experiments carefully and, furthermore, developed a quantum theory that nicely explains the observed photocurrent spectra. This paper describes how the strongly coupled light-matter states couple with the electron transport and would become a seminal paper in this field. Therefore, I strongly recommend the publication of this paper in Nature Communications."

A2.0 We thank the reviewers for his positive assessment of our work.

R2.1. "The QCD sample used in this study has 11 repetitions of the QCD unit and the quantum wells are heavily doped. In the analysis, the authors assume that, when a voltage is applied to the sample, the electric field is uniform over an entire structure. However, it is known that, when multiple quantum wells are doped, as in the cases of quantum well infrared photodetector structures, so-called high-field domains are formed. This effect strongly modifies the internal field distribution. I wonder how the authors avoid or take into account the effect of high-field domains in their experiments."

A2.1. We are aware that heavily doped multiple quantum well structures can give rise to high field domains and field non-uniformity in general. In particular, this regime has been observed in QWIP detectors under strong illumination and high applied bias [1]. Owing to the photovoltaic nature of our QCD our measurements were performed with low power (we estimated the FTIR output to be a few microwatts) and low bias (max: 0.35 V). The IV characteristic that we measure are smooth (see Figure R2.1) and we did not observe "kinks" in the differential resistance up to this voltage, which are typically a sign for domain formation [2] (note that signs of a non-linear behavior appears at higher voltages ~0.5 V). Furthermore, all bias dependent measurements were modelled with the same linewidth parameters as in the 0V case see Fig. 3a of the main text and Fig. SI.6 in the supporting material). A strongly non-uniform distribution would probably result in resonance broadening for increasing bias, which we did not observe. The shape of photocurrent spectra in our device should be particularly sensitive to such inhomogeneity, as they change significantly with the energy difference between level 2 and 3.

Note that even if we do not necessarily assume a uniform electric field over the entire structure, our data indicates that the energy shift of the transition towards the extraction will be the same in all periods of the structure. Indeed, as the referee indicates, the field distribution is inhomogeneous within each period: when a bias is applied, we can refer to the case of quantum cascade lasers, where the electric field distribution is still not uniform in the single period (as the highly doped QW absorbs most of the potential drop) but remains periodic along the structure [3]. In that case our conclusions are not altered, as the main spectral features of the photocurrent arise only from the energy difference between level 2 and 3. Our data on Figure 3a indicates that this should be in that case, in the regimes of low bias that we have reported.

Figure R2.1 IV and differential resistance at 5K in a MESA structure. No clear sign of negative differential resistance was observed within the range of bias applied to the structure (green area).

R2.2. (very minor comment)

1) line 215: the authors say “at negative bias”. However, it was confusing to me, because I thought the authors control the position of level 3 with respect to the intersubband transition. In this case, I think “at positive bias” seems to be more appropriate.

2) line 96 (typo?): “with”  “which”

A2.2. We thank the reviewer for the attention to details.

We have corrected the indicates typos.

We do agree with the referee that the sign of the applied bias could be misleading. Our choice is due to the fabrication of double-metal devices which involves wafer bonding and consecutive flipping of the structure, with a reverse of the sequence of the epitaxial growth. The bottom contact in the structure thus corresponds to the end of the growth (or the end of the cascade). Therefore, as we applied bias through that contact, we applied a negative potential. This is exactly the same as the application of a positive potential to the top of the structure.

To avoid confusion, we have changed the notation as suggested by the reviewer.

- [1] Schneider, H., et al. "Domain pinning in GaAs/AlGaAs quantum well infrared photodetectors." *Applied Physics Letters* 88.5 (2006): 051114.
- [2] K. K. Choi, et al., “Periodic negative conductance by sequential resonant tunneling through an expanding high-field superlattice domain”, *Phys. Rev. B* **35**, (1987): 4172(R)
- [3] Beere, H. E., et al. "MBE growth of terahertz quantum cascade lasers." *Journal of crystal growth* 278.1-4 (2005): 756-764.

Referee 3

R3.0 “...The paper is well written and describes significant progress in the field of infrared photodetection under ultrastrong, non-perturbative coupling with the resonator mode. The results are convincing and the methodology sound. I recommend publication in *Nature Communications* with some minor revisions as indicated below.”

A3.0 We thank the reviewers for his positive assessment of our work.

R3.1. “Some words regarding the influence of the number N_p of QCD periods would be important. In particular, the sentence in lines 162-166 regarding the interpretation of G_t is contradictory: If one

electron is transferred from one period to the next one, $1/N_p$ electrons will circulate in the external circuit. Therefore, "the number of electrons circulating in the read-out circuit for each absorbed photon" is not the same as "the probability of having an electron transferred to the next period of the quantum cascade once it has been photoexcited". The value of these two terms differs by the factor N_p . Please check.

The small peak value of ca. 4% in Fig. 3a may suggest that the factor $1/N_p$ is already contained implicitly in G_t (in this case, the factor $1/N_p$ should be added to the definition of G_t rather than to eq. (2))."

A3.1. We thank the referee about this important remark, that help us revise our expression of the responsivity and include the correct dependence of the factor $1/N_p$ in Eq.(2).

In our model, the current is provided by the quantity eN_3/τ_{34} where N_3 is the population of level 3 in a single period. Indeed, in a multiperiod structure, we assume that there is a continuity of the current on average so that an electron leaving a period to the next one (or the right contact) is replaced with another electron entering from a previous period (or left contact). This is the "Eulerian" picture of the transport in QCD [1].

However, the absorption efficiency of the structure η in Eq. (3) corresponds to the total absorption of the structure. Since we are far from the saturation regime, all light-induced populations are proportional to the incident power; therefore the total populations on level 3 from all periods $N_p N_3$ is proportional to the total absorption efficiency, $N_p N_3 \sim \eta$. Thus the photocurrent is $I_{ph} = eN_3/\tau_{34} \sim \eta/N_p$, and hence Eq.(2) should be corrected by a factor $1/N_p$.

However, this change does not affect our results, and in particular the spectral dependence of the photocurrent. In our original submission, as described in the Supplementary material we used the time t_{34} as a fitting parameter which allowed to recover the absolute value of the responsivity that we estimate from our experiments. Previously, we used $\tau_{34} = 7$ ps to reproduce the experimental results, but now we set it to $\tau_{34} = 0.3$ ps, which coincidentally is closer to the values found in literature [2][3] and better follows the expected LO phonon electron scattering time. Consequently, we updated the value of the tunnel gain G_t ; however, the coherent gain G_H is not affected. Indeed, it has been expressed only from quantities (energy position of the extraction level, scattering rates...) that are related to a single period and do not depend on the total populations/coherences. Its expression is therefore unaltered.

In order to amend these issues, the following explanation have been added in the main text:

Page 5: Eq. (2) has been corrected with a $1/N_p$ factor;

Page 5: just after Eq.(2): "the absorption efficiency" has been replaced with "the total absorption efficiency";

Page 5, last paragraph: the phrase "In conventional QCD theory [32,33], G_t is the only factor present in the expression of the responsivity and can be seen as the number of electrons circulating in the read-out circuit for each absorbed photon," has been removed. Instead, we have added the following explanations:

"Since $0 < G_t < 1$, it can be interpreted as the probability of having an electron transferred to the next period of the quantum cascade once it has been photoexcited. In the conventional QCD theory [32,33, 34], G_t is referred as the extraction probability, " p_e ", and is the only quantity entering the responsivity besides the quantum efficiency and the number of periods N_p ."

Page 5 and 6, a paragraph has been added:

“In our model, we have adopted the “Eulerian” picture of the transport in QCD as described in [34], where there is continuity of the current, that is an electron leaving one period is replaced by another electron entering from a contact or the adjacent period. In the expression of the current $I_{ph} = eN_3/\tau_{34}$, N_3 is the population of a single period of the QCD. However, the absorption efficiency $\eta(\hbar\omega)$ entering Eq.(2) is the one for the whole absorbing region. Our model described the Supplementary material actually provides the total population of all extractor levels, N_3N_p which in linear regime is proportional to the total absorption $\eta(\hbar\omega)$. Hence in Eq.(2) we introduce the factor $1/N_p$; in the conventional QCD theory the detector gain, that is the number of electrons circulating in the read-out circuit for each absorbed photon, is thus provided by G_t/N_p [Error! Reference source not found.,Error! Reference source not found.,34].”

Reference list: [1] has been added.

Figures: we refer to G_t as an extraction probability instead of “tunnel gain”.

Supplementary material, an explanation on the $1/N_p$ factor has been added in the end of 1.1.

R.3.2. *In Fig. 3b, the peak value of G_H is about the same as N_p (presumably $N_p=11$ was chosen because of previous results). Does theory predict such a limit (i.e., that coherence is limited by the size of the device)?*

A.3.2. In Figure 3b of the main text, we present the spectral shape of the coherent gain for different values of the extractor energy. The peak of the function increases monotonically with higher energy and can exceed the number of periods. For example, the peak value corresponding to the extractor with $\hbar\omega_{13} = 160$ meV is close to 20, and higher values can be achieved for even higher energies. However, it is important to note that this calculation assumes a constant scattering rate γ_{13} and neglects any contribution from the finite depth of the QW. We made this choice to ensure a more consistent study of the coherent gain dependence on the energy transition. While the theory does not predict a limit for the peak value of G_H , experimental limitations will eventually restrict this increase. Indeed, as the level 3 becomes closer to the edge of the QW, all levels lying in the continuum must be included in the model, as electrons can escape as well by scattering in the continuum rather than tunneling from 3. The model thus must be generalized by taking the infinity of levels in the continuum, which corresponds to QWIP, rather than a QCD structure; we believe that this will lead to a natural upper bound of the G_H . In our original submission, this was indicated as note [N1], we have inserted the text of this note in the main text,

Page 6, third phrase above equation (3):” Note that the maximal value of $\hbar\omega_{13}$ is naturally limited by the height of the AlGaAs barriers, while the level 3 is reaching the conduction band edge of AlGaAs the levels in the continuum must be included in the model”.

R.3.3. *Fig. 1b, upper panel: Most likely, the zero of the photocurrent at about 118meV is caused by a sign reversal. This might be attributed to the 1-5 transition leading to transport in the opposite direction. Perhaps the authors could briefly comment on this.*

A.3.3. We appreciate the reviewer's suggestion. Indeed, the zero of the photocurrent indicates that there are different path that electrons could undergo which would result in a negative contribution at low energies. Here we comment on what we believe might be the most plausible explanation, but we did not included it in the main text as it would require further work to quantify this hypothesis.

The 1-5 transition should primarily occur towards the right side (resulting in positive current), as the wavefunction overlap favors that direction. Therefore, in our calculation, we have assumed a “primary path” of electron flow as described in Figure S1. The referee is however right, and there is a possible backflow as indicated in the Figure R3.1 below which creates a counter current. In the figure below, we

present the simulated band structure for 2 periods. (Note that the Poisson self-consistent potential was not included to reduce the simulation time.) For instance, after an absorption an electron promoted on level 2, and then is scattered/tunnels toward level 5', or undergoes a diagonal transition from 1 to 5'. Then from there it may be scattered to the level 1'. This mechanism can be accounted for by including additional terms in the equations for the populations. There could be also a coherent effect driven by the coherence 1-5', similar to the one described in the main text. However, this process seems to be more favorable for "low" energy photons, where the excitation rate of the coherence $1 \rightarrow 3$ is lower. This could explain the fact that negative photocurrent is observed at lower photon energies. However, we could not provide a plausible explanation for the backscattering from 5' to 1', which in our design seems rather unlikely owe to the vanishing overlap of the corresponding wavefunctions; therefore we chose not to comment this ideas in the current submission.

Figure R3.1: Band diagram of the structure under investigation. Two period were simulated to show an alternative path for electrons that would yield negative contribution to the photocurrent.

- [1] Delga, Alexandre. "Quantum cascade detectors: A review." *Mid-infrared Optoelectronics* (2020): 337-377.
- [2] Luo, Tengfei, et al. "Gallium arsenide thermal conductivity and optical phonon relaxation times from first-principles calculations." *Europhysics Letters* 101.1 (2013): 16001.
- [3] Zhou, Jin-Jian, and Marco Bernardi. "Ab initio electron mobility and polar phonon scattering in GaAs." *Physical Review B* 94.20 (2016): 201201.

REVIEWERS' COMMENTS

Reviewer #1 (Remarks to the Author):

The authors have answered the comments of the reviewer in a satisfactory manner and the paper can therefore be published.

Reviewer #2 (Remarks to the Author):

In this paper, the authors investigate how the light-matter strongly coupled states are converted to a photocurrent by using a quantum cascade detector (QCD) structure. The quantum well in the QCD structure was very heavily doped. As a result, the intersubband plasmon mode is strongly coupled with the carefully designed cavity mode and forms a strongly coupled state. The authors observed an enhancement in the responsivity spectrum of the QCD structure when the light-matter coupling is strong. They developed a theory that takes into account the coupling with microcavity photons, collective electronic effects, and tunnel coupling with the extractor states, and clearly showed the coherence between the ground state (1) and the extractor state (3) plays a very important role in the photocurrent generation process, although the oscillator strength between 1 and 3 is negligibly small. This is the process in which electrons are directly transferred from state 1 to state 3 assisted by the collective polarization due to photon absorption. In other words, the enhancement in the responsivity results from the rectification of the fast oscillating optical field and quantum coherences, i.e., the conversion from ac optical fields to a dc current. The authors performed their experiments carefully and, furthermore, developed a quantum theory that nicely explains the observed photocurrent spectra. This paper describes how the strongly coupled light-matter states couple with the electron transport and would become an important paper in this field.

In the first round revision process, the authors appropriately revised the manuscript. Therefore, I strongly recommend the publication of this paper in Nature Communications.

Reviewer #3 (Remarks to the Author):

In their revised manuscript together with the rebuttal letter, the authors are taking account of all criticisms and questions raised by the referees.

Just the newly added sentence in lines 208-211 should be rewritten: (i) The "=" sign should either be omitted or some value should be given after it. (ii) Reaching the AlGaAs bandedge is probably not a good criterion for the necessity to include the continuum. On one hand, level #3 should be located significantly below the AlGaAs conduction bandedge to make sure that discrete states are a good approximation. On the other hand, the model might still work reasonably well when the highest level has developed into an above-barrier resonance. (iii) Please check the grammar.

Otherwise I recommend to publish the article as is.

Dear editor and referees,

Please find enclosed our point by point answer to the final comments and questions from the referees. We reported in italic the comments from the referees, and our answers follow in normal fonts. A minor remark from referee 3 required a modification to the main text, which is highlighted in green in the revised file.

Sincerely,

Francesco Pisani, Yanko Todorov and the authors.

Referee 1

The authors have answered the comments of the reviewer in a satisfactory manner and the paper can therefore be published.

Answer 1

We thank the referee for the positive assessment of our work.

Referee 2

In this paper, the authors investigate how the light-matter strongly coupled states are converted to a photocurrent by using a quantum cascade detector (QCD) structure. The quantum well in the QCD structure was very heavily doped. As a result, the intersubband plasmon mode is strongly coupled with the carefully designed cavity mode and forms a strongly coupled state. The authors observed an enhancement in the responsivity spectrum of the QCD structure when the light-matter coupling is strong. They developed a theory that takes into account the coupling with microcavity photons, collective electronic effects, and tunnel coupling with the extractor states, and clearly showed the coherence between the ground state (1) and the extractor state (3) plays a very important role in the photocurrent generation process, although the oscillator strength between 1 and 3 is negligibly small. This is the process in which electrons are directly transferred from state 1 to state 3 assisted by the collective polarization due to photon absorption. In other words, the enhancement in the responsivity results from the rectification of the fast oscillating optical field and quantum coherences, i.e., the conversion from ac optical fields to a dc current. The authors performed their experiments carefully and, furthermore, developed a quantum theory that nicely explains the observed photocurrent spectra. This paper describes how the strongly coupled light-matter states couple with the electron transport and would become an important paper in this field.

In the first round revision process, the authors appropriately revised the manuscript. Therefore, I strongly recommend the publication of this paper in Nature Communications.

Answer 2

We thank the referee for the nice summary and positive comments regarding our work.

Referee 3

In their revised manuscript together with the rebuttal letter, the authors are taking account of all criticisms and questions raised by the referees. Just the newly added sentence in lines 208-211 should be rewritten: (i) The "=" sign should either be omitted or some value should be given after it. (ii) Reaching the AlGaAs bandedge is probably not a good criterion for the necessity to include the continuum. On one hand, level #3 should be located significantly below the AlGaAs conduction bandedge to make sure that discrete states are a good approximation. On the other hand, the model might still work reasonably well when the highest level

has developed into an above-barrier resonance. (iii) Please check the grammar. Otherwise I recommend to publish the article as is.

Answer 3

We thank the referee for noticing the typo, we removed the “=” sign. The phrase has been also changed to: “Note that the maximum value of $\hbar\omega_{13}$ is naturally limited by the height of the AlGaAs barriers, indeed the level 3 must be sufficiently below the barrier edge to avoid coupling with the continuum.”; to take into account the referee’s suggestion.

We thank for the support to the publication of our work.